# Impact of layered behavioral, socio-economic and school-based interventions on selected behavioral and biomarker indicators among adolescent girls and young women in Uganda

Joseph K. B. Matovu[1,2]*, John Baptist Bwanika[1], Irene Murungi[3], Jacqueline K. Kyambadde[3], Ntombekhaya Matsha-Carpentier[4], Saman Zamani[4], Rhoda K. Wanyenze[1]

1 Makerere University School of Public Health, Kampala, Uganda, 2 Busitema University Faculty of Health Sciences, Mbale, Uganda, 3 The AIDS Support Organization, Kampala, Uganda, 4 The Global Fund to Fight AIDS, Tuberculosis and Malaria, Geneva, Switzerland

* jmatovu@musph.ac.ug

## Abstract

Globally, adolescent girls and young women (AGYW) continue to be at an elevated risk of HIV infection. We assessed the impact of layered behavioral, socio-economic and school-based interventions on selected behavioral and biomarker indicators among AGYW in Uganda. We used data from two serial cross-sectional surveys conducted in 14 (eight intervention and six comparison) districts in 2018 ($n = 8,236$) and 2023 ($n = 5,449$). Between 2019 and 2023, AGYW in the intervention districts received social and behavioral change communication plus either socio-economic support, vocational skills-based training or educational subsidies, as appropriate. AGYW in the comparison districts were not exposed to these interventions. We collected data on eight behavioral and two biomarker (HIV, syphilis) indicators. Exposure to AGYW interventions was defined as participation in or receipt of at least one intervention and expressed as a percentage. Impact was determined using a difference-in-difference approach with the observed net effect assumed to represent the overall impact of the interventions. We compared the effect of the interventions between exposed and unexposed AGYW and between intervention and non-intervention districts or schools. Data were analyzed using STATA, version 16.0. Half of the AGYW were in school while 50–70% were single/never married. Overall exposure to AGYW interventions was low; ranging from 4.4% to 43% depending on the type of intervention. Exposure to the interventions had a small net effect (0.7% to 14%) on almost all behavioral indicators and HIV prevalence was much higher among exposed than unexposed AGYW (1.56% *vs.* 0.94%). Syphilis prevalence was much lower among exposed than unexposed AGYW (0.26% vs. 0.82%) with an overall marginal net decline of 1.0% in the intervention versus non-intervention districts or schools. These findings suggest a need for a critical re-appraisal of the design

**Data availability statement:** All data can be accessed with permission from the Makerere University School of Public Health Research and Ethics Committee (sphrecadmin@musph. ac.ug), upon reasonable request.

**Funding:** This study was implemented as part of a sub-grant awarded to Makerere University School of Public Health by The AIDS Support Organization (grant#: UGA-C-TASO), with funding support from The Global Fund to Fight AIDS, Tuberculosis and Malaria (The Global Fund). The funders had no role in study design, data collection and analysis, decision to publish, or preparation of the manuscript. The views and opinions expressed in this article are the responsibility of the authors and do not necessarily reflect the views and opinions of TASO or The Global Fund.

**Competing interests:** The authors have declared that no competing interests exist.

and implementation of interventions targeting AGYW in order to achieve the desired changes in behavioral and biomarker indicators.

## Introduction

Adolescent girls and young women (AGYW) in Eastern and Southern Africa (ESA) continue to be at a heightened risk of HIV infection, teenage pregnancy and intimate partner violence [1–4]. Although the ESA region accounts for less than 10% of the world's population, it accounts for 53% of all new HIV infections [5]. Within this region, 15% of all new infections occur among adolescents aged 10–19 years, with 83% of adolescent infections occurring in girls [6]. Using national HIV population-based household surveys conducted between 2008 and 2017 in South Africa, Mabaso et al. [7] found that HIV prevalence remained higher among AGYW (ranging between 4.2-5.7%) than among their male counterparts (where HIV prevalence ranged between 1.8-4.5%) over the eight-year period [7]. Results from the most recent 5th Botswana AIDS Impact Survey show that while HIV prevalence was 1.6% among adolescent boys aged 15–19 years, HIV prevalence among their female counterparts was 2.7%. Similarly, among those aged 20–24 years, HIV prevalence was 2.7% among young men as opposed to 6.7% among young women [8]. In Uganda, a recent population-based HIV impact assessment found that HIV prevalence among 15–19-year-old adolescent girls was 1.7% compared to 0.2% among adolescent boys of the same age, with similarly higher HIV prevalence levels among young women aged 20–24 years (4.2%) compared to 1.6% among young men of the same age [9]. A recent meta-analysis found that HIV incidence among female adolescents exceeded that of male adolescents across ten high-prevalence African countries [10]. Moreover, over half of all HIV infections among adolescent girls globally occur in just 10 countries in ESA, including Uganda [10,11]. Long-standing gender inequalities, discrimination and poverty deny many women and adolescent girls' economic autonomy, deprive them of control over their sexual lives, and expose them to ongoing HIV risk [7,12].

Studies show a clear link between intimate partner violence (IPV) and HIV [13–19]. Using demographic and health survey data from 27 countries, Wado et al. [4] found that experience of any intimate partner violence (physical, sexual and physical or sexual violence) ranged from 6.5% in Comoros to 43.3% in Gabon, with a median of 25.2%. IPV can impact HIV susceptibility if women feel unable or have difficulty negotiating safe sex practices with the abusive partner [13]. Women who experienced IPV may have been sexually assaulted by a risky male partner, which directly affects their HIV susceptibility [13,14]. Adolescent girls in physically abusive relationships were three times more likely to become pregnant than non-abused girls [15]. In a study that assessed the association between sexual violence and unintended pregnancy among adolescent girls and young women in South Africa, Ajayi & Ezegbe [16] found that unintended pregnancy was higher among survivors of sexual violence (54.4%) compared to those who never experienced sexual abuse (34.3%). Adolescent mothers who are in violent relationships may find it difficult to refuse sexual activity or

to negotiate contraceptive use with an agitated partner. A study of more than 500 adolescent mothers from Texas found that those who experienced physical abuse within three months after delivery were nearly twice as likely to have a repeat pregnancy within 24 months [17].

In addition to HIV and IPV, adolescent birth rates (ABR) continue to remain high, particularly in sub-Saharan Africa. For instance, while the number of annual births per 1,000 adolescents among 15–19-year-old girls has decreased from 64.5 births per 1000 women in 2000 to 41.3 births per 1000 women in 2023, rates of change have been uneven in different regions of the world with the slowest declines observed in the Latin American and Caribbean (LAC) and sub-Saharan Africa (SSA) regions [18]. Besides, while the global adolescent birth rate for girls aged 10–14 years has also declined from 3.3 to 1.5 births per 1000 adolescent girls over the same period, adolescent birth rates for girls aged 10–14 years remain higher in SSA (at 4.4 births per 1000 adolescent girls) and LAC (at 2.3 births per 1000 adolescent girls) than in any other regions of the world [18,19]. A host of factors continue to drive teenage pregnancy rates in SSA including child marriages, low contraceptive use, child sexual abuse and intimate partner violence [20]. These observations suggest a need for combination interventions to simultaneously prevent child marriages, adolescent pregnancies and childbearing in this part of the world.

Evidence shows that exposure to multiple HIV prevention interventions (i.e., layered interventions) can help to reduce HIV risk among AGYW [21–25]. For instance, studies show that girls who received a combination of school support, pre-exposure prophylaxis (PrEP) and violence prevention were more likely to report consistent condom use than those who received school support alone [21]. Similarly, girls who received cash transfers delivered along with combination HIV prevention interventions were more likely to report reduced HIV-risk behavior and reduced sexually transmitted infections than those who received either intervention alone [22–23]. In-school HIV prevention interventions, linking comprehensive sexuality education with sexual and reproductive health services, can improve adolescent-parent communication, HIV knowledge and attitudes, self-efficacy in condom use, a decrease in risky sexual behavior, and decrease early and unintended pregnancies among AGYW [24]. Other promising interventions include promoting gender-equitable norms and addressing gender-based violence which have been demonstrated to improve HIV testing uptake and HIV knowledge and attitudes; out-of-school HIV prevention education, making health services more adolescent friendly, and skills training programs [25]. Thus, implementation of multi-level interventions is crucial for improved targeting of a multitude of risk behaviors that continue to put AGYW at risk of the triple burden of HIV, teenage pregnancy and intimate partner violence. However, few studies have examined the impact of multi-level interventions to generate data needed to inform their scale-up to reach more vulnerable girls.

This paper reports on the findings from an evaluation of the impact of layered behavioral, socio-economic and school-based interventions on selected behavioral and biomarker indicators among adolescent girls and young women in Uganda. The impact evaluation was conducted to: a) determine if interventions targeting AGYW had any impact on selected behavioral and biomarker indicators and b) inform the scale-up of HIV, teenage pregnancy and intimate partner violence prevention interventions to reach more vulnerable girls in Uganda. The AGYW interventions include those that targeted in-school girls (implemented by the Ministry of Education and Sports) and those that targeted out-of-school girls (implemented through The AIDS Support Organization [TASO], Ministry of Health, and Ministry of Gender, Labor and Social Development).

## Methods

### Ethical statement

The two (2018 and 2023) surveys were all approved by the Research and Ethics Committee at Makerere University School of Public Health (Protocol#593 [2018] and SPH-2023–436 [2023]) and cleared by the Uganda National Council for Science and Technology (UNCST; Protocol# SS4678 [2018] and HS2943ES [2023]). Because the surveys included girls

below the age of consent in Uganda (i.e., before age 18), we obtained written informed consent from the parents/guardians (including for in-school AGYW) and written assent from the girls themselves. Otherwise, all AGYW aged 18–24 years provided their own written informed consent in line with the UNCST guidelines.

## Study design, sites and population

This analysis uses data from 14 high HIV-burden districts that were surveyed as part of two serial cross-sectional surveys conducted in July 2018 and July 2023. Details about the 2018 survey have been published elsewhere [26]. In brief, with support from the Global Fund, the 2018 survey was conducted among in- and out-of-school AGYW aged 10–24 years, resident in 20 high HIV-burden districts. The 20 districts were selected purposively (based on their HIV prevalence at the time) and included Kalangala, Nakasongola, Kiboga, Buikwe, Jinja, Buyende, Kaliro, Bugiri, Tororo, Mbale, Bukwo, Busia, Hoima, Kyankwanzi, Kasese, Kisoro, Amolatar, Otuke, Amuru, and Kitgum. The 2018 survey served to provide the baseline estimates upon which the impact of interventions targeting AGYW aged 10–24 years would be measured. Following the 2018 survey, layered AGYW interventions (comprising social and behavioral change communication [SBCC], educational subsidies, socio-economic support and vocational skills-based training) were initiated in eight of the 20 surveyed districts in 2019 with ongoing funding support until 2023. However, in the other 12 districts, only SBCC interventions were implemented. The 2023 survey used the same methodology as the 2018 survey and targeted in- and out-of-school AGYW aged 10–24 years, resident in 14 of the original 20 districts surveyed in 2018. The 14 districts surveyed in 2023 included eight districts (i.e., Buikwe, Busia, Jinja, Nakasongola, Bukwo, Hoima, Mbale and Tororo) that received layered interventions and six districts (Amolatar, Bugiri, Kalangala, Kasese, Kisoro and Otuke) that received SBCC-only interventions (see S1 Text for details on the interventions). It is important to note that six of the original 20 districts that were surveyed in 2018 were not resurveyed in 2023 due to logistical and administrative considerations.

## Overview of the AGYW program

Please see S1 Text for a detailed description of the AGYW program. In brief, between 2019 and 2023, the AGYW program was implemented in 20 high HIV-burden districts with the goal of addressing issues affecting AGYW's access to health, social and economic services in order to reduce their levels of vulnerability to HIV and other sexually transmitted infections. AGYW interventions included: SBCC, vocational skills-based training; enterprise development assistance (girls with viable businesses were provided with economic support/capitation grants to expand their businesses); second-chance, non-formal education for girls who dropped out of school (girls were trained in skills-based courses that did not require them to return to a formal school environment) through empowerment clubs and innovation camps; and the *Sinovuyo Teen* program with a focus on improving communication between parents and their adolescent daughters, among others.

Interventions targeting out-of-school AGYW were implemented by The AIDS Support Organization (TASO) with technical support from the Ministry of Health and the Ministry of Gender, Labor and Social Development. These interventions were implemented through sub-recipient organizations including Baylor Uganda; Program for Accessible health, Communication and Education (PACE); Uganda Development and Health Associates (UDHA), and TASO-Bukwo. On the other hand, interventions targeting in-school AGYW were implemented directly through the Ministry of Education and Sports (MoES). The school-based interventions were implemented in the same districts as the out-of-school interventions, except that these interventions targeted in-school rather than out-of-school girls. No similar interventions targeted in- or out-of-school AGYW in the comparison districts or schools. If AGYW in the comparison districts received any of these interventions, this should have been either through government-supported health facilities or programs (e.g., through universal primary or secondary education that is provided to all school-going pupils and students) or through any other implementing partners that operated in those districts. This was done to ensure that a clear difference in outcomes could be determined between AGYW in the intervention versus non-intervention districts or schools.

Given that the factors contributing to the triple burden of HIV, teenage pregnancy and gender-based violence among AGYW may be complex with multiple interacting components, it was necessary that TASO defined the mechanisms and assumptions through which change was expected to occur after the girls were exposed to the interventions. These mechanisms and assumptions informed the design of the AGYW program's Theory of Change. The term "Theory of Change" is used in this context to refer to a set of mechanisms and assumptions put together to describe how SBCC, educational subsidies, skills-based training, and economic support interventions targeting AGYW were expected bring about specific long-term outcomes through a logical sequence of strategic actions [27]. Based on the AGYW program's Theory of Change (Fig 1), the interventions were expected to improve girls' level of economic independence and safer sex negotiation skills, and with these skills, girls would be able to avoid engaging in high-risk behaviors. The reduction in high-risk behaviors was assumed to result in reduced new HIV infections among exposed relative to unexposed girls, with impact expected to be visible at both individual and district levels.

## Study population

The two surveys (2018 and 2023) targeted the same study population: adolescent girls and young women (AGYW) aged 10–24 years who were in- or out of school, studying in schools or living within the surveyed districts, respectively.

## Sample size

The 2018 survey had a sample size of 8,236 AGYW while the 2023 survey used a sample size of 5,449 AGYW. The sample sizes were estimated using the same sample size determination method. The sample size was calculated per selected district using population sizes for each year preceding the survey (i.e., 2017 for the 2018 survey and 2022 for the 2023 survey). These populations were obtained based on the 2014 Uganda National Population and Housing Census [28] using district population growth rates (average of 3.0% reported in the census report). The target population size (females aged 10–14, 15–19 and 20–24 years in and out of school) was generated by considering an estimated proportion of 33.6% of

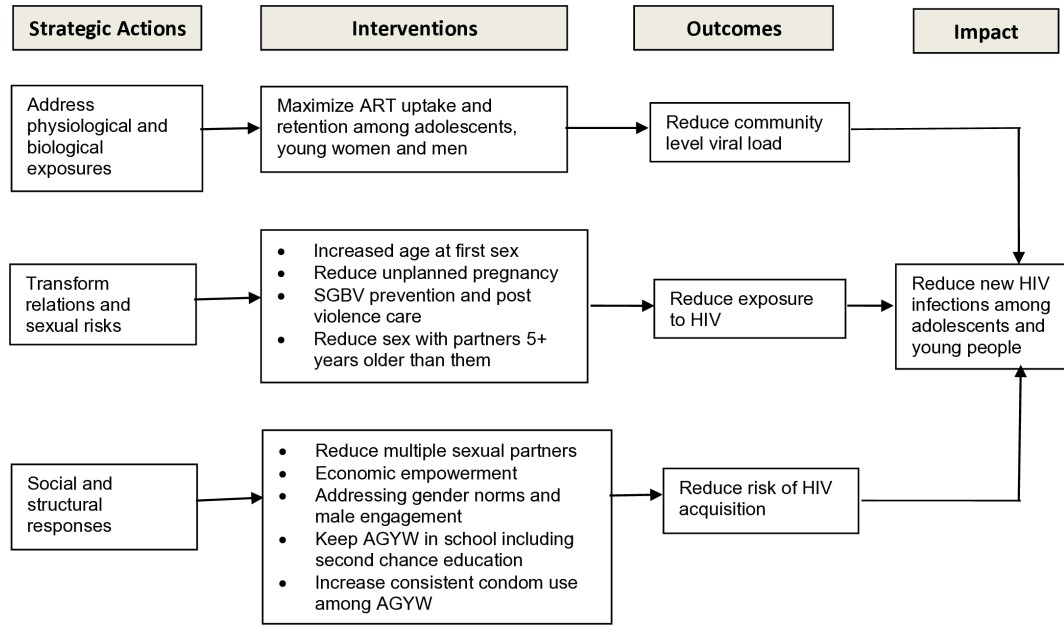

**Fig 1. Theory of change for the AGYW program.**

females in the age group 10–24 who left school, 61.6% in school and 4.7% who had never been in school, and these proportions were assumed to be equal for all districts and age groups. To obtain the sample size per district and age group, we used the formula for sample surveys shown below [29].

$$n = \frac{Z_{1-\alpha/2}^2 P(1-P)N}{d^2 N + Z_{1-\alpha/2}^2 P(1-P)}$$

Where 'P' was the anticipated population proportion of the study outcome (here we used HIV prevalence of 1% among the target population, based on the results from the baseline study), 'N' was the target population size and 'Z' is z-score set to 1.96 at $\alpha = 5\%$ significance level.

## Sampling procedures

The sampling methodology for the 2018 and 2023 surveys is similar. The sampling methodology for the 2018 survey has been described previously [26]. In brief, in-school girls were surveyed at school (during school time) while out-of-school girls were surveyed at household level (in their respective villages of residence). In consultation with the Ministry of Education and Sports, a list of all schools where AGYW interventions were implemented in each district was obtained. We checked for schools surveyed at baseline (2018) and grouped them into intervention (where Global Fund-supported AGYW activities took place) and non-intervention schools (no Global Fund-supported AGYW activities). Five intervention and five non-intervention schools were selected per district. A list of pupils/students was obtained from the school administration in each of the 10 schools. Approximately 20 AGYW were interviewed per school, matched on age.

To select the villages, we retrieved the list of villages (from our archives) where the baseline survey was conducted in 2018 and grouped them into intervention vs. non- intervention villages, based on whether, subsequently, there were any Global Fund-supported AGYW interventions in that village. All the intervention villages that were surveyed in 2018 were included in the 2023 survey plus a sub-sample of other villages where interventions were being implemented, even if they were not surveyed in 2018. This list constituted the list of intervention villages. We then selected a comparable list of non-intervention villages (i.e., villages that were surveyed in 2018 but received no AGYW interventions) to constitute the list of non-intervention villages. Within the 20 villages per district, a list of households with out-of-school AGYW was generated in consultation with the local leaders. From each selected village, approximately 10 AGYW, matched on age, were interviewed.

## Data collection procedures and methods

The 2018 survey began on 24/07/2018 and ended on 30/08/2018 while the 2023 survey began on 03/07/2023 and ended on 25/08/2023. In the 2018 survey, data were collected in 233 villages and 80 schools while in the 2023 survey, data were collected in 280 villages and 140 schools. In the 2023 survey, data were collected from the same villages and schools as in the 2018 survey to aid in the computation of the difference-in-difference estimates during the assessment of the impact of the interventions on the behavioral and biomarker indicators (see *'Measurement of variables'* below). For each survey, we used structured questionnaires to collect socio-demographic and behavioral data from eligible AGYW aged 10–24 years. To be eligible for interview, a girl had to be resident in the selected village (if out-of-school) or studying within the selected school (for in-school AGYW). All girls provided written informed consent prior to participation in the study.

The 2018 survey used paper-based questionnaires that were then entered into the computer while the 2023 survey used digital data collection methods. In the 2023 survey, a digital questionnaire was preloaded on mobile devices such as phones and tablets, and data collected using the *KoboCollect* mobile application. The questions used in both surveys were based on those obtained from the Uganda Demographic and Health Survey Questionnaire and additional questions were generated by the investigators based on gaps identified in the literature. A few questions were added to the 2023

survey. All questionnaires were designed with appropriate skip patterns and validation criteria to enable accurate data capture. However, in the 2023 survey, research assistants were required to upload the data daily to the cloud server for quality control purposes. Upon submission, data were downloaded into the Microsoft Excel program for cleaning. Data cleaning involved the removal of unwanted or duplicate observations from the dataset (de-duplication), fixing structural errors such as typos, or incorrect capitalization, filtering unwanted outliers, missing data, and validation.

## Blood sample collection and field HIV and syphilis testing

Upon completion of the survey, AGYW were requested to voluntarily provide a blood sample for HIV and syphilis testing. Our laboratory procedures have been described previously [26]. In summary, all AGYW who participated in the survey were subjected to standard pre- and post-test HIV and syphilis counseling and tested by trained Laboratory Technicians and Nurse Counselors who were part of the survey team in each district. AGYW who tested positive for syphilis were referred for syphilis treatment at the nearby health facilities while those who tested HIV-positive were referred for immediate linkage to HIV care at the same health facilities, as per the National HIV Testing Services (HTS) Policy and Implementation Guidelines [30]. Respondents were informed that they could opt for both HIV and syphilis tests; only one of the two tests, or none of the tests and that they could decide, for each test, whether or not they wanted the results to be given to them.

Blood was collected in EDTA vacutainer tubes (with anticoagulant). All rapid tests (HIV and syphilis) were performed in the field. All preliminary HIV-positive results and 5% of HIV-negative results were shipped to the Central Public Health Laboratory (CPHL) hub for onward delivery to the main CPHL head office in Kampala for confirmatory testing and quality control. Samples were transported to CPHL at 2-8°C. Laboratory Technicians placed labels with the same study identification number onto each vacutainer tube as well as on each Field Test Results Form. HIV quality control testing was done at CPHL following the National HTS Policy and Implementation Guidelines [30]. HIV testing was done using *Determine* (Alere, USA), *STAT-PAK* (Chembio Diagnostics, USA) and *SD-Bioline* (Standard Diagnostics, Korea), in that order. All samples that showed inconclusive HIV result either from the field or at CPHL and those that were not concordant after screening at CPHL were subjected to a DNA PCR test to confirm the true results. Syphilis quality control testing was done using SD Bioline TP 3.0 kit and all positive samples were subjected to antibody titration using Rapid Plasma Reagin (RPR) test to evaluate activeness of infection.

All HIV-positive results were handled as per the National HTS Policy and Implementation Guidelines [30]. For minors, before disclosing results, the counsellor assessed if the parent or guardian was willing to discuss HIV and the test results with the child openly. Disclosure was done with the person with whom the child feels most comfortable. For children who could not clearly understand the results (e.g., those aged 10–11 years), the parent or guardian was fully involved. However, children who were 12 years or older were given their individual results after proper counselling, with the involvement of parents or guardians, where appropriate. Children below 12 years of age were given results only with the consent of parents or guardians and with proper counselling. All AGYW confirmed HIV positive were referred to the nearest health facility for enrolment into HIV care and linked to peer-leaders that would support them with follow-up for retention in care and treatment. In addition, those who tested HIV negative, but were syphilis positive were referred to the nearest health facility for further management.

## Measurement of variables

The following variables are used in the reporting of findings. This sub-section presents a summary description on how they were defined and measured.

a) ***Exposure to AGYW interventions***: Exposure to the interventions was defined as a situation where a girl received SBCC messages, participated in the SBCC activities or received scholastic materials such as exercise books and

menstrual pads among those who were in school. We used the following parameters to define exposure to the interventions:

i. Participation in social and behaviour change communication (SBCC) activities designed for AGYW in the last 4 years [*Games (netball, football, volleyball, etc.); music, dance, and drama; community sports events; community outreaches/meetings; and essay competitions*)]

ii. Receiving messages that required action regarding any of the following behaviors [*Screening for STIs; testing for HIV; enrolling in HIV care; using family planning methods; reducing number of sexual partners; and consistent condom use with sexual partner*]

iii. Receiving support or participation in any of the following socio-economic or empowerment activities among AGYW [*Second chance education; vocational skills-based training; enterprise development assistance (EDA); empowerment clubs; dialogue meetings; and thematic radio talk shows*]

iv. Receiving any of the following items, distributed by the Ministry of Education and Sports at school [*Free exercise books; underwear; half-petty; and menstrual pads*]

It is important to note that all AGYW received exposure to SBCC activities plus any of the other interventions, as appropriate. For instance, a girl could receive SBCC and/or socio-economic support or empowerment or SBCC with vocational skills-based training. We computed a composite variable to determine the proportion of AGYW who were exposed to at least one intervention activity within each category, and this was interpreted as "participation in or receipt of at least one intervention component". We then computed a composite of these individual composite variables to determine an aggregate measure of exposure or non-exposure to the interventions. That is, each of the four areas mentioned above had a composite variable created to measure participation or receipt of at least one intervention component, and these composites were put together to determine an 'aggregate composite' of the different composite variables, which was categorized as a binary variable – i.e., exposure versus non-exposure to the interventions.

b) ***Socio-economic status***: Responses on household possessions were used to create an index representing the socio-economic status (SES) of the AGYW interviewed. The list of household assets probed for included a radio, television, bicycle, motorcycle, mobile phone, fixed phone, refrigerator, electricity, indoor bathroom, car, solar power, agricultural land and livestock. To construct the SES index, each household item was assigned a weight ascertained through principal components analysis. Then, the scores were standardized in relation to a standard normal distribution with a mean of zero and a standard deviation of one. For everyone, the scores on household possessions were then summed up. Then, individuals were ranked and sub-divided into wealth tertiles, depending on their scores, with each tertile containing 30% of the participants. The SES index, generated as above, was then applied to several analyses to understand the differences in social and health behaviour of respondents that can be attributed to SES.

c) ***Impact of AGYW interventions***: The impact of AGYW interventions on the primary outcomes (IPV, teenage pregnancy, and HIV prevalence) was determined by examining changes in behavioral and biomarker indicators. Eight behavioral indicators (i.e., *comprehensive HIV knowledge; intimate partner violence (physical, sexual); proportion reporting 2+sexual partners in the past 12 months; condom use at last sex; current use of modern contraceptive methods; HIV testing uptake in the past 12 months; proportion reporting that they had their first pregnancy before age 18, and proportion who had sexual debut before age 15*) were assessed in addition to two biomarker indicators (HIV and syphilis) using data from two serial surveys (2018 and 2023). Since the interventions were delivered at district or school level, we computed the difference between the proportions reported in the 2018 and 2023 surveys (i.e., proportion reported in 2023 minus the one reported in the 2018) separately for intervention versus non-intervention districts/schools, and computed a difference-in-difference estimate for each outcome. This estimate was assumed to represent

the impact of the interventions on each primary outcome. Finally, we computed the difference between the indicators as reported among girls who were exposed to the interventions versus those that were not exposed to the interventions to determine if exposure to the interventions led to improvements in behavioral or biomarker indicators when compared to non-exposure to these interventions. As already mentioned, a girl was considered exposed to the interventions if she participated in at least one of the intervention components that were delivered to the girls by the implementing organizations.

### Data analysis

In the intervention and non-intervention districts/schools, program reach was computed as a proportion of those reached with the interventions divided by the number of those interviewed for each respective category. In the computation of biomarker indicators, population size weights were applied using the *svy:* command, as appropriate, for estimates generation. All the selected behavioral and biomarker indicators were computed as proportions, stratified by intervention status (intervention *vs.* non-intervention). This level of analysis was done for each survey year, with estimates generated for both intervention and non-intervention districts. Because changes in behavioral and biomarker indicators occurred in both the intervention and non-intervention districts, we computed the difference between the proportions reported in 2023 and those reported in 2018 for both the intervention and non-intervention districts, and then subtracted the difference in the non-intervention districts from the difference in the intervention districts to determine a difference-in-difference estimate. The difference-in-difference estimate represents the observed changes in behavioral or biomarker indicators that can be attributed to the intervention. In general, as part of data analysis, we aimed to: a) compare behavioral and biomarker data by exposure status (i.e., exposed vs. non-exposed AGYW); b) assess changes in selected behavioral and biomarker indicators in the *intervention* vs. *non-intervention* districts and schools (2018 *vs.* 2023); and c) determine if there was a marked impact of the interventions on the selected indicators. We computed the 95% confidence intervals around the different biomarker estimates using the *proportion* command in STATA. All data analyses were conducted using STATA version 16.0.

## Results

### Population characteristics

Of the girls surveyed in 2018 and 2023, half were in school; 60–70% were aged 18 and 24 years; nearly two-thirds were single/never married while nearly one-third were in the lowest wealth tertile. Across the surveys, the characteristics of the girls in the intervention versus non-intervention districts were kept constant to ensure comparability of findings (Table 1).

The reporting of results has been divided into two parts: a) exposure to AGYW interventions and c) impact of AGYW interventions on selected behavioral and biomarker indicators. The impact of the interventions has been further divided into: i) impact among exposed *versus* non-exposed AGYW; ii) impact among AGYW in the intervention *versus* non-intervention districts, and iii) impact among AGYW in the intervention *versus* non-intervention schools.

**Exposure to AGYW interventions.** A detailed analysis of AGYW's exposure to the different interventions is provided in S2 Text. In summary, exposure to the interventions was low, ranging from 4.4% to 43%, based on the type of intervention under consideration. Specifically, only 4.4% (n = 137) of girls in the intervention and 0.08% (n = 2) of girls in the non-intervention districts reported that they participated in any SBCC activity; 14.6% (n = 458) of girls in the intervention districts and 5.4% (n = 127) of girls in the non-intervention districts reported that they received any message that required them to take action; while 15.8% (n = 495) of girls in the intervention districts and 7.9% (n = 185) of girls in the non-intervention districts reported that they received support or participated in any socio-economic or economic empowerment activity. Receipt of education materials was at 43% (n = 676) in the intervention versus 16.8% (n = 195) in the non-intervention districts. The most reported items included menstrual pads at 40.4% (n = 636) and 12.4% (n = 144)

**Table 1. Characteristics of the study population by intervention status.**

| Characteristic | Intervention districts | | Non-intervention districts | |
|---|---|---|---|---|
| | 2018 survey (n = 3,333, %) | 2023 survey (n = 3,125, %) | 2018 survey (n = 2,438, %) | 2023 survey (n = 2,332, %) |
| **Schooling status** | | | | |
| In-school | 1,666 (50.0) | 1571 (50.3) | 1,229 (50.4) | 1,164 (49.9) |
| Out-of-school | 1,667 (50.0) | 1554 (49.7) | 1,209 (49.6) | 1,168 (50.1) |
| **Age-group** | | | | |
| 10-14 | 495 (14.9) | 562 (18.0) | 370 (15.2) | 384 (16.5) |
| 15-17 | 415 (12.5) | 603 (19.3) | 396 (16.2) | 469 (20.1) |
| 18-19 | 843 (25.3) | 742 (23.7) | 832 (34.1) | 535 (22.9) |
| 20-24 | 1,580 (47.4) | 1,218 (39.0) | 840 (34.5) | 944 (40.5) |
| **Marital status** | | | | |
| Single/never married | 1,722 (51.7) | 2,066 (66.1) | 1,752 (71.9) | 1,486 (63.7) |
| In relationship but not married | 876 (26.3) | 522 (16.7) | 227 (9.3) | 476 (20.4) |
| Currently married | 582 (17.5) | 388 (12.4) | 361 (14.8) | 218 (9.3) |
| Divorced/widowed/separated | 153 (4.6) | 149 (4.8) | 98 (4.0) | 152 (6.5) |
| **Wealth tertile** | | | | |
| Lowest | 945 (28.4) | 1,029 (32.9) | 947 (38.8) | 820 (35.2) |
| Middle | 1,074 (32.2) | 1,018 (32.6) | 781 (32.0) | 769 (33.0) |
| Highest | 1,314 (39.4) | 1,078 (34.5) | 710 (29.1) | 743 (31.9) |

in the intervention and non-intervention districts and free exercise books at 37.9% (n = 596) and 10.1% (n = 117) in the intervention and non-intervention districts, respectively.

Table 2 shows the frequency of girls being provided with information on how to avoid teenage pregnancy while at school during the school term, within the intervention and non-intervention districts, and by exposure versus non-exposure status. Overall, less than half of the girls (47.9%, n = 1,308) reported that they often or very often received information on how to avoid teenage pregnancy during the school term. Half of the girls in the intervention districts (50.2%, n = 790) compared to 44.5% (n = 518) of girls in the non-intervention districts reported that they were often or very often provided with information on how to avoid pregnancy during the school term. On the other hand, a higher proportion of unexposed girls (56.3%, n = 864) reported that they were often or very often provided with information on how to avoid pregnancy during the school term when compared to exposed girls (37.0%, n = 444), possibly due to the work of other HIV service providers other than the sub-recipients.

**Table 2. Frequency of being provided with information on how to avoid teenage pregnancy while at school, during the school term.**

| Characteristics | Very often, n (%) | Often, n (%) | Less often, n (%) | Not at all, n (%) | Total, n (%) |
|---|---|---|---|---|---|
| **Overall** | **420 (15.4%)** | **888 (32.5%)** | **683 (25%)** | **744 (27.2%)** | **2735 (100%)** |
| **District intervention status** | | | | | |
| Intervention | 263 (16.7%) | 527 (33.5%) | 425 (27.1%) | 356 (22.7%) | 1571 (100%) |
| Non-Intervention | 157 (13.5%) | 361 (31%) | 258 (22.2%) | 388 (33.3%) | 1164 (100%) |
| **Exposure status** | | | | | |
| Exposed | 122 (10.2%) | 322 (26.8%) | 277 (23.1%) | 480 (40%) | 1201 (100%) |
| Not Exposed | 298 (19.4%) | 566 (36.9%) | 406 (26.5%) | 264 (17.2%) | 1534 (100%) |

Table 3 shows the frequency of access to HIV information among in-school girls during the school term. Overall, less than half (46.8%, n = 1,280) of in-school girls reported that they were often or very often provided with information on how to avoid HIV infection during the school term. Nearly half of the girls in the intervention districts (49.7%, n = 780) compared to 42.9% (n = 500) of girls in the non-intervention districts reported that they were often or very often provided with information on how to avoid HIV infection during the school term. Up to 22.0% (n = 345) of girls in the intervention and 33.4% (n = 389) of girls in the non-intervention districts did not receive any information at all on how to avoid HIV infection during the school term.

**Impact of AGYW interventions on selected behavioral and biomarker indicators.** The following sub-sections present a summative overview of the impact of AGYW interventions on behavioral and biomarker indicators at individual, district and school-level. The findings are stratified by exposure *vs.* non-exposure and intervention *vs.* non-intervention districts or schools.

*Impact of interventions on behavioral and biomarker indicators among exposed vs. unexposed AGYW:* Table 4 presents a summative overview of the impact of exposure to the interventions on behavioral indicators among exposed and unexposed girls, based on the 2023 survey data. As noted, exposure to AGYW interventions had small but positive impacts on almost all behavioral indicators except on the percentage of AGYW reporting 2 + partners in the past 12 months. The greatest impact was seen for comprehensive HIV knowledge with 50.8% (n = 1,341) in the exposed *vs.* 37.1% (n = 1,045) in the non-exposed girls (a 13.7% difference) and condom use at last sex with 29.0% (n = 428) of exposed AGYW reporting condom use at last sex *vs.* 21.1% (n = 257) among unexposed girls (a 7.9% difference). The rest of the changes were basically small, ranging from a reduction of 1.7% to an increase of 3.3% in some indicators.

Table 5 shows the impact of exposure to the interventions on biomarker indicators among exposed relative to unexposed girls. HIV prevalence was higher in the exposed group (1.56%; 95% confidence interval [95%CI]: 0.73, 3.34) than

**Table 3. Frequency of receiving information on how to avoid HIV infection at school, during the school term.**

| Characteristics | Very often, n (%) | Often, n (%) | Less often, n (%) | Not at all, n (%) | Total, n (%) |
|---|---|---|---|---|---|
| **Overall** | **397 (14.5%)** | **883 (32.3%)** | **721 (26.4%)** | **734 (26.8%)** | **2735 (100%)** |
| **District intervention status** | | | | | |
| Intervention | 257 (16.4%) | 523 (33.3%) | 446 (28.4%) | 345 (22%) | 1571 (100%) |
| Non-Intervention | 140 (12%) | 360 (30.9%) | 275 (23.6%) | 389 (33.4%) | 1164 (100%) |
| **Exposure status** | | | | | |
| Exposed | 105 (8.7%) | 325 (27.1%) | 278 (23.1%) | 493 (41%) | 1201 (100%) |
| Not Exposed | 292 (9.0%) | 558 (6.4%) | 443 (8.9%) | 241 (5.7%) | 1534 (0.0%) |

**Table 4. Impact of AGYW interventions on behavioral indicators by exposure status.**

| Indicator | EXPOSED | NON-EXPOSED | DIFFERENCE |
|---|---|---|---|
| Comprehensive HIV knowledge | 50.8% (N = 2,639) | 37.1% (N = 2,818) | 13.7% |
| Intimate partner violence (physical) | 2.8% (N = 865) | 4.5% (N = 4,598) | -1.7% |
| Intimate partner violence (sexual) | 2.8% (N = 865) | 4.2% (N = 4,598) | -1.4% |
| Proportion reporting 2 + sexual partners in the past 12 months | 13.0% (N = 2,639) | 11.1% (N = 2,824) | 1.9% |
| Condom use at last sex | 29.0% (N = 1,475) | 21.1% (N = 1,226) | 7.9% |
| Current use of modern family planning methods | 45.7% (N = 674) | 42.4% (N = 680) | 3.3% |
| HIV testing uptake (past 12 months) | 61.5% (N = 1,304) | 58.5% (N = 1,099) | 3.0% |
| Proportion of AGYW that got pregnant for the first time before age 18 | 14.2% (N = 2,639) | 15.3% (N = 2,824) | -1.1% |
| Proportion of AGYW who had sexual debut before age 15 | 9.6% (N = 2,639) | 10.3% (N = 2,818) | -0.7% |

**Table 5. Weighted HIV and syphilis prevalence by exposure to AGYW interventions.**

| Indicator | Exposed to AGYW (%, 95%CI) | Unexposed to AGYW (%, 95%CI) |
|---|---|---|
| **Weighted HIV prevalence** | | |
| **Overall** | **1.56 (0.73, 3.34)** | **0.94 (0.49, 1.78)** |
| **Schooling status** | | |
| In-school | 0.96 (0.37, 2.47) | 0.49 (0.00, 2.9) |
| Out-of-school | 2.37 (0.77, 7.09) | 1.18 (0.56, 2.46) |
| **Age-group** | | |
| 10-14 | 0.19 (0.00, 1.77) | 1.1 (0.30, 3.9) |
| 15-17 | 0.79 (0.18, 3.42) | 0.71 (0.14, 3.53) |
| 18-19 | 0.99 (0.35, 2.76) | 0.92 (0.29, 2.84) |
| 20-24 | 2.72 (1.35, 5.4) | 1.06 (0.25, 4.46) |
| **Weighted syphilis prevalence** | | |
| **Overall** | 0.26 (0.10, 0.65) | 0.82 (0.37, 1.78) |
| **Schooling status** | | |
| In-school | 0.18 (0.00, 0.72) | 0.0 |
| Out-of-school | 0.36 (0.11, 1.13) | 1.26 (0.55, 2.84) |
| **Age-group** | | |
| 10-14 | 0.0 | 0.0 |
| 15-17 | 0.12 (0.00, 1.2) | 0.0 |
| 18-19 | 0.0 | 1.59 (0.37, 6.66) |
| 20-24 | 0.59 (0.23, 1.53) | (0.37, 1.78) |

in the non-exposed group (0.94%; 95%CI: 0.49, 1.78), but syphilis prevalence was lower in the exposed than the non-exposed group (0.26% [95%CI: 0.1, 0.65] *vs.* 0.82% [95%CI: 0.37, 1.78]).

***Impact of AGYW interventions on behavioral and biomarker indicators in the intervention vs. non-intervention districts:*** Table 6 shows the impact of the interventions on the behavioral indicators in the intervention and non-intervention districts. When we compared the changes in both types of districts at the two time points, we observed impact in only three behavioral indicators: *current use of modern family planning* (increased by 8.6% in the intervention vs. 4.4%

**Table 6. Impact on behavioral indicators in the intervention *vs.* non-intervention districts.**

| Indicator | Intervention Districts | | | Non-intervention Districts | | | Difference in Difference |
|---|---|---|---|---|---|---|---|
| | 2018 | 2023 | % Diff* | 2018 | 2023 | % Diff* | |
| Comprehensive HIV knowledge | 46.40% | 44.7% | -1.70% | 44.20% | 42.40% | -1.80% | 0.10% |
| Intimate physical partner violence in the past 12 months | 14.90% | 4.00% | -10.90% | 18.20% | 4.10% | -14.10% | 3.20% |
| Intimate sexual partner violence in the past 12 months | 6.20% | 3.70% | -2.50% | 9.80% | 4.50% | -5.30% | 2.80% |
| Reported 2＋sexual partners in the past 12 months | 12.40% | 12.00% | -0.40% | 24.30% | 13.40% | -10.90% | 10.50% |
| Condom use at last sex | 10.10% | 25.50% | 15.40% | 6.90% | 22.80% | 15.90% | -0.50% |
| **Current use of modern family planning methods** | **36.20%** | **44.80%** | **8.60%** | **38.60%** | **43.00%** | **4.40%** | **4.20%** |
| Tested for HIV in the past 12 months | 72.50% | 61.50% | -11.00% | 69.00% | 58.30% | -10.70% | -0.30% |
| **Teenage pregnancy (first pregnancy before age 18)** | **16.50%** | **10.9%** | **-5.60%** | **11.90%** | **13.1%** | **1.20%** | **-6.80%** |
| **Sexual debut (before 15 years of age)** | **16.10%** | **10.00%** | **-6.10%** | **12.50%** | **8.50%** | **-4.00%** | **-2.10%** |

*\*Diff, Difference*

increase in the non-intervention districts), *proportion reporting teenage pregnancy* (-5.6% reduction in the intervention *vs.*+1.2% increase in the non-intervention districts), and *the proportion of AGYW reporting sexual debut before the age of 15* (-6.1% reduction in the intervention *vs.* -4.0% reduction in the non-intervention districts). However, the difference in the three behavioral indicators that can be attributed to the district-level AGYW interventions was minimal, ranging from -6.8% to +4.2%. The interventions did not impact on comprehensive knowledge of HIV, intimate physical and sexual violence, proportion of AGYW reporting 2+sexual partners in the past 12 months, condom use at last sex, and the proportion reporting HIV testing uptake in the past 12 months. Actually, in some of these indicators, the reduction was higher in the non-intervention than in the intervention districts.

Table 7 below shows the changes in biomarker indicators in the intervention vs. non-intervention districts between 2018 and 2023. Overall, following exposure to the interventions, HIV prevalence increased in both the intervention (from 1.0% to 1.4%) and non-intervention districts (from 1.1% to 1.3%) during this period. In both intervention and non-intervention districts, HIV prevalence was higher among out-of-school than in-school girls and increased with increasing age. Among out-of-school AGYW specifically, HIV prevalence was similar in both the intervention and non-intervention districts, and increased from 1.8% in 2018 to 1.9% in 2023 in both types of districts. Across age-groups, HIV prevalence increased between 2018 and 2023, with the largest increase seen in those aged 20–24 years.

Unlike HIV prevalence, syphilis prevalence decreased more markedly in the intervention districts from 1.4% to 0.4% and minimally in the non-intervention districts from 1.17% to 1.16% between 2018 and 2023. As it was the case with HIV prevalence, syphilis prevalence was higher among out-of-school AGYW and those aged 20–24 years. In the intervention districts, syphilis prevalence decreased more markedly in the young adolescents from 0.94% to 0.0% among those aged 10–14 years and from 0.68% to 0.0% in those aged 15–17 years. Even among those aged 20–24 years, where syphilis

Table 7. Impact of AGYW interventions on biomarker indicators in the intervention *vs.* non-intervention districts.

| Indicator | Intervention districts | | Non-intervention districts | |
|---|---|---|---|---|
| | 2018 (%, 95%CI) | 2023 (%, 95%CI) | 2018 (%, 95%CI) | 2023 (%, 95%CI) |
| **Weighted HIV prevalence** | | | | |
| **Overall** | **1.00 (0.61, 1.40)** | **1.40 (0.70, 2.63)** | **1.10 (0.59, 1.61)** | **1.30 (0.65, 2.57)** |
| **Schooling status** | | | | |
| In-school | 0.49 (0.00, 1.60) | 0.86 (0.33, 2.2) | 0.59 (0.30, 1.00) | 0.69 (0.30, 1.60) |
| Out-of-school | 1.79 (0.09, 3.49) | 1.88 (0.71, 4.87) | 1.78 (1.08, 2.48) | 1.89 (0.66, 5.32) |
| **Age-group** | | | | |
| 10-14 | 0.54 (0.03, 1.04) | 0.68 (0.22, 2.09) | 0.60 (0.21, 1.01) | 0.94 (0.23, 3.73) |
| 15-17 | 0.68 (0.00, 5.94) | 0.77 (0.18, 3.18) | 0.78 (0.00, 2.94) | 0.68 (0.00, 5.94) |
| 18-19 | 0.67 (0.00, 4.76) | 0.97 (0.43, 2.16) | 0.82 (0.16, 1.48) | 0.67 (0.16, 2.76) |
| 20-24 | 1.57 (1.01, 2.59) | 2.21 (1.06, 4.56) | 2.03 (1.31, 3.00) | 2.27 (1.31, 3.90) |
| **Weighted syphilis prevalence** | | | | |
| **Overall** | 1.40 (0.92, 1.88) | 0.40 (0.21, 0.89) | 1.17 (0.54, 2.51) | 1.16 (0.84, 1.61) |
| **Schooling status** | | | | |
| In-school | 0.69 (0.30, 1.6) | 0.14 (0.00, 0.57) | 0.12 (0.00, 0.99) | 1.02 (0.54, 1.92) |
| Out-of-school | 1.89 (0.66, 5.32) | 0.73 (0.34, 1.56) | 2.04 (1.28, 3.23) | 1.30 (0.76, 2.24) |
| **Age-group** | | | | |
| 10-14 | 0.94 (0.23, 3.73) | 0 | 0.90 (0.00, 1.73) | 0.87 (0.15, 4.83) |
| 15-17 | 0.68 (0.00, 5.94) | 0 | 0.88 (0.00, 2.04) | 1.59 (0.86, 2.94) |
| 18-19 | 0.67 (0.16, 2.76) | 0.44 (0.11, 1.70) | 1.03 (0.33, 3.17) | 0.73 (0.16, 3.21) |
| 20-24 | 2.27 (1.31, 3.90) | 0.74 (0.40, 1.38) | 1.83 (0.95, 3.50) | 0.70, 2.15) |

prevalence remains higher, there was a marked reduction in syphilis prevalence from 2.27% to 0.74% in the intervention districts, and from 1.83% to 1.23% in the non-intervention districts. Thus, AGYW interventions impacted on syphilis prevalence but there was no impact on HIV prevalence which increased between 2018 and 2023.

***Impact of AGYW interventions on behavioral and biomarker indicators in the intervention vs. non-intervention schools:*** Table 8 provides a summative overview of the impact of AGYW interventions on selected behavioral indicators in the intervention versus non-intervention schools. There was evidence of intervention effect in only three behavioral indicators: *intimate physical violence* (decreased by 8.5% in the intervention schools compared to 7.0% in the non-intervention schools); *intimate sexual violence* (reduction by 1.4% vs. 1.2%), and *condom use at last sex* (increased by 24.8% vs. 23.3%). However, the difference in behavioral indicators that can be attributed to school-based AGYW interventions was very minimal, ranging from -1.5% to +1.5%.

Conversely, there were better outcomes in the non-intervention than intervention schools regarding reductions in teenage pregnancy, reductions in the proportion of girls reporting sexual debut before age 15, current use of modern family planning methods, proportion reporting 2+sexual partners in the past 12 months, and comprehensive knowledge of HIV. HIV prevalence remained stable at 0.6% between 2018 and 2023 in the intervention schools but dropped from 2% to 0.8% in the non-intervention schools (Table 9). Syphilis prevalence declined by 1.0% in the intervention schools from 1.2% to 0.2% but increased from 0.4% to 0.7% in the non-intervention schools.

## Discussion

Our findings show that: a) exposed girls had better outcomes than unexposed girls across almost all behavioral indicators but the difference between exposed and unexposed girls was very modest with no observed impact on HIV prevalence and b) the impact of AGYW interventions on behavioral indicators at district or school level was very modest with no observed impact on HIV prevalence. While we cannot fully explain the marginal effect of the interventions on the behavioral indicators, several reasons may help to explain these observations: a) the intensity and coverage of interventions may not have been at the level necessary to cause a marked change in behavior (for instance only 47% and 48% of in-school AGYW often/very often received information on how to avoid HIV or teenage pregnancy during the school term, respectively); b) the period of intervention exposure may have been too short to allow for behavior change to occur (for instance, some interventions lasted for only 10 days); and c) the mode of implementation of social and behavioral change communication interventions, which was more of a crowd-puller, may have affected the ability of the program implementers to apply sound health communication principles, including use of target-specific, audience-segmented communication [31]. In general, while study findings appear to be in the direction envisioned through the program's Theory of Change, the modest intervention effects indicate that the

**Table 8. Impact of AGYW interventions on behavioral indicators in the intervention *vs.* non-intervention schools.**

| Indicator | Intervention Schools | | | Non-intervention Schools | | | Difference in Difference |
|---|---|---|---|---|---|---|---|
| | 2018 | 2023 | % Diff | 2018 | 2023 | % Diff | |
| Comprehensive knowledge of HIV | 44.80% | 46.00% | 1.20% | 46.60% | 48.40% | 1.80% | 0.60% |
| **Intimate physical partner violence in the past 12 months** | **9.30%** | **0.80%** | **-8.50%** | **7.60%** | **0.60%** | **-7.00%** | **-1.50%** |
| **Intimate sexual partner violence in the past 12 months** | **2.70%** | **1.30%** | **-1.40%** | **2.90%** | **1.70%** | **-1.20%** | **0.20%** |
| Prop reporting 2+sexual partners in the past 12 months | 20.90% | 7.20% | -13.70% | 24.60% | 10.30% | -14.30% | 0.60% |
| **Condom use at last sex** | **8.10%** | **32.90%** | **24.80%** | **9.10%** | **32.40%** | **23.30%** | **1.50%** |
| Current use of modern family planning methods | 36.20% | 40.90% | 4.70% | 36.20% | 41.90% | 5.70% | -1.0% |
| Tested for HIV in the past 12 months | 60.40% | 49.40% | -11.00% | 63.60% | 55.70% | -7.90% | -3.10% |
| Teenage pregnancy (first pregnancy before age 18) | 0.50% | 0.90% | 0.40% | 0.80% | 1.70% | 0.90% | -0.50% |
| Sexual debut (before 15 years of age) | 7.10% | 6.30% | -0.80% | 8.00% | 4.50% | -3.50% | 2.70% |

**Table 9.  Impact of AGYW interventions on biomarker indicators in the intervention *vs.* non-intervention schools.**

| Indicator | Intervention schools | | Non-intervention schools | |
|---|---|---|---|---|
| | 2018 (%, 95%CI) | 2023 (%, 95%CI) | 2018 (%, 95%CI) | 2023 (%, 95%CI) |
| **Weighted HIV prevalence** | | | | |
| **Overall** | **0.6 (0.12, 3.26)** | **0.6 (0.20, 2.01)** | **2.0 (1.10, 3.00)** | **0.8 (0.40, 1.60)** |
| **Age-group** | | | | |
| 10-14 | 2.14 (0.64, 6.95) | 0.27 (0.00, 2.92) | 0.90 (0.00, 1.73) | 0.87 (0.15, 4.83) |
| 15-17 | 0 | 0.77 (0.15, 3.89) | 0.88 (0.00, 2.04) | 1.59 (0.86, 2.94) |
| 18-19 | 0 | 0 | 1.03 (0.33, 3.17) | 0.73 (0.16, 3.21) |
| 20-24 | 0.65 (0.00, 4.81) | 1.4 (0.26, 7.15) | 1.83 (0.95, 3.50) | 1.23 (0.70, 2.15) |
| **Weighted syphilis prevalence** | | | | |
| **Overall** | 1.20 (0.56, 2.63) | 0.20 (0.00, 1.48) | 0.40 (0.11, 0.69) | 0.70 (0.26, 1.71) |
| **Age-group** | | | | |
| 10-14 | 3.22 (1.31, 7.68) | 0 | 0.12 (0.00,1.24) | 1.2 (0.30,4.65) |
| 15-17 | 0 | 0 | 0.59 (0.00, 4.68) | 0.88 (0.23, 3.34) |
| 18-19 | 0 | 0 | 0.61 (0.13, 2.74) | 0.49 (0.00, 2.90) |
| 20-24 | 1.50 (0.30, 7.07) | 0.66 (0.00, 7.48) | 0.42 (0. 20, 0.27) | 0.55 (0.20, 1.51) |

assumptions underlying the AGYW's Theory of Change were not realized. In future, it will be important for program implementers to pay a close attention to not only the intensity of intervention delivery but also to the fidelity of implementation and duration of exposure to improve the impact of AGYW interventions on the selected behavioral and biomarker indicators.

Our findings show that there was no observed impact of the AGYW interventions on HIV prevalence. HIV prevalence increased in both the intervention and non-intervention districts, with a more marked increase in the intervention districts. We do not know why HIV prevalence increased in both sets of districts (and more so, in the intervention districts). It is likely that the increase in HIV prevalence is an artificial one resulting from sampling bias where we might have interviewed more girls who were already living with HIV (due to pre-existing infections) in the 2023 survey than in the 2018 survey but this is just a possibility. It is also likely that some girls may have acquired HIV after exposure to the interventions (due to engaging in HIV risk behavior); resulting in an increase in HIV prevalence. For instance, the proportion reporting condom use at last sex increased only marginally from 10.1% to 25.5% while the proportion reporting 2+sexual partners in the past 12 months remained stable at 12% between the two surveys. Alternatively, it is also likely that the interventions may not have been implemented at the level of intensity, coverage and duration needed to impact on HIV incidence, thereby resulting in an increase in HIV prevalence. Further research is warranted to fully understand what caused an increase in HIV prevalence between the two surveys, and more so, within the intervention districts. Nevertheless, the lack of impact on HIV indicators after exposure to AGYW interventions has been reported elsewhere. In 2021, using AGYW data from the DREAMS program in Kenya and South Africa, Birdthistle et al. [32] reported that while declines in HIV incidence were observed in the period preceding the implementation of the DREAMS interventions, these declines were not sustained in the first three years of DREAMS implementation. The team concluded that the previous declines in HIV incidence could have been driven by earlier and ongoing investments in HIV testing and treatment rather than the ongoing DREAMS interventions. Similarly, in an analysis of data from a South African cohort study, Mthiyane et al. [33] found no evidence that HSV-2 or HIV incidence or transmissible HIV (i.e., detectable HIV viral load) were lower among AGYW exposed to the DREAMS interventions than those that were not, after two years of exposure to the DREAMS interventions. Collectively, these findings suggest that AGYW interventions are yet to yield the desired changes in biomarker indicators in countries where they have been implemented.

Study findings show that the interventions had a modest impact on the targeted behavioral indicators. Specifically, interventions had a small but positive effect on current use of modern contraception, first pregnancy before the age of 18 years, sexual debut before the age of 15, condom use at last sex and intimate (physical, sexual) violence in the past 12 months. Our findings are consistent with previous studies that show modest or no impact of interventions on behavioral indicators among AGYW [34–36] although more marked changes in behaviors following exposure to the interventions have been reported in other studies [37,38]. The lack of impact of interventions targeting AGYW on behavioral outcomes may be partly due to the way they were designed or implemented [39,40] or partly due to the COVID-19 interruptions in 2020 and 2021. For instance, in a study that assessed family planning service disruptions in the first two years of the COVID-19 pandemic in seven countries (including Uganda), Karp et al. [41] reported reduced health facility service hours, high provider absenteeism, and more irregular contraceptive services in three of the six countries studied. Similar disruptions have been reported elsewhere [42,43]. However, although we did not document the impact of COVID-19 interruptions on the AGYW studied, evidence from Uganda suggest that the impact of these interruptions could have been minimal given that they were smaller in magnitude and lasted for a short duration of time [44]. Besides, the Ugandan Ministry of Health put in place policy guidance to ensure continued availability of essential services, including during community outreaches [45,46]. Thus, while the COVID-19 pandemic affected access to health services, it is likely that its impact on the attainment of the tracked indicators may have been limited. Nevertheless, the fact that some interventions show some level of impact on behavioral indicators is essential in strengthening their implementation (e.g., through increased fidelity, intensity of implementation, and better targeting of the most vulnerable girls) to improve their ability to yield better outcomes.

Our findings have programmatic and research implications. For instance, while we did not assess implementation fidelity quantitatively, we observed that some interventions were not implemented as initially designed due to the need to meet intervention targets within the shortest time possible. For instance, most social and behavior change communication activities were implemented in the format of large, one-size-fits-all approaches rather than through targeted small group meetings that would take the project team more time to reach set targets. Such deviations can alter the attainment of the desired outcomes, leading to erroneous conclusions about the impact of the interventions. Thus, future interventionists should ensure that there is a high level of implementation fidelity across all intervention areas to ensure similarity of exposure to these interventions. In addition, there is a need to ensure that all interventions are linked to defined primary outcomes. Also, during our interaction with the project team, we were informed of cases where out-of-school AGYW were trained in vocational skilling but were not provided with a start-up kit. Such AGYW ended up not utilizing the skills acquired. Any future implementation of such interventions should be accompanied with a defined primary outcome, e.g., percentage of AGYW who utilize the vocational skills acquired to set up an income generating activity, or percentage of in-school AGYW who have been retained in school because they received menstruation pads that enabled them to attend school daily even during their menstruation days. Finally, future research is warranted to understand the relative contribution of the different interventions separately in order to determine the intervention synergies that are essential to maximally improve the attainment of the desired HIV outcomes [47].

## Study limitations and strengths

Our study had several limitations and strengths. First, our analysis of intervention impact did not examine the role of specific interventions due to limited coverage of these interventions among AGYW targeted by these interventions. Thus, we were not able to tease out the relative contribution of the different interventions on the primary outcome. Secondly, given that the interventions that we evaluated were implemented between 2019 and 2023; it is likely that their implementation was affected by the effects of the COVID-19 lockdown in 2020 and 2021 [48]. However, since we did not document the effects of the COVID-19 lockdown on the implementation of the interventions, further research is necessary to justify this limitation. Thirdly, besides the interventions implemented with the support of the Global Fund in Uganda, there

were other interventions that targeted AGYW in the "comparison districts" during the same intervention implementation period, thereby diluting the likely intervention effect in the intervention communities. We tried to frame the questions in the questionnaire in such a way as to help AGYW to recall if the interventions that they received were from one of the Global Fund-supported implementing partners. For instance, after assessing if the respondents had participated in a community event (e.g., community sports event), we would ask respondents if the event was organized by the designated implementing partner in the area. This approach might have helped to reduce the problem of misclassifying AGYW as having received the interventions from a Global Fund-supported implementing partner when they actually did not receive them from that partner. Nevertheless, despite these limitations, we believe that this study is one among a few studies that have been conducted to assess the impact of combined AGYW interventions on both behavioral and biomarker indicators. Besides, our study had a large sample size that enabled us to make precise statistical estimates with a high level of confidence.

## Conclusion

Our findings show that, overall, layered behavioral, socio-economic and school-based interviews had a modest impact on behavioral indicators but no impact on HIV prevalence. Specifically, at the individual level, exposed girls had better outcomes than unexposed girls across almost all behavioral indicators but the difference between the outcomes was modest, making it difficult to attribute this difference to the impact of the interventions. Besides, HIV prevalence was much higher among exposed than unexposed girls, suggesting that exposure to the interventions did not impact on HIV prevalence. On the other hand, while AGYW in the intervention districts had better outcomes on some behavioral indicators, the impact of these interventions on the behavioral indicators was very modest across all tracked indicators. Finally, HIV prevalence increased in both the intervention and non-intervention districts, with a more marked increase in the intervention districts, suggesting that the interventions did not have any impact on HIV prevalence. Collectively, these findings call for a need to critically re-examine the design, implementation modalities (coverage, fidelity) and intervention packages for both in- and out-of-school AGYW in order to address areas that still hamper the attainment of the desired impact on behavioral and biomarker impact indicators.

## Supporting information

**S1 Text. Detailed description of the Global Fund intervention package for AGYW.**
(DOCX)

**S2 Text. Detailed analysis on exposure to AGYW interventions.**
(DOCX)

## Author contributions

**Conceptualization:** Joseph KB Matovu, Rhoda K. Wanyenze.

**Data curation:** John Baptist Bwanika.

**Formal analysis:** John Baptist Bwanika.

**Funding acquisition:** Joseph KB Matovu, Irene Murungi, Jacqueline K. Kyambadde, Ntombekhaya Matsha-Carpentier, Saman Zamani, Rhoda K. Wanyenze.

**Investigation:** Joseph KB Matovu.

**Methodology:** Joseph KB Matovu, John Baptist Bwanika, Irene Murungi, Jacqueline K. Kyambadde, Ntombekhaya Matsha-Carpentier, Saman Zamani, Rhoda K. Wanyenze.

**Project administration:** Jacqueline K. Kyambadde, Saman Zamani.

**Supervision:** Joseph KB Matovu, Irene Murungi, Ntombekhaya Matsha-Carpentier, Saman Zamani, Rhoda K. Wanyenze.

**Validation:** Joseph KB Matovu, John Baptist Bwanika, Ntombekhaya Matsha-Carpentier, Saman Zamani, Rhoda K. Wanyenze.

**Visualization:** Irene Murungi, Jacqueline K. Kyambadde.

**Writing – original draft:** Joseph KB Matovu, John Baptist Bwanika, Irene Murungi, Jacqueline K. Kyambadde, Ntombekhaya Matsha-Carpentier, Saman Zamani, Rhoda K. Wanyenze.

**Writing – review & editing:** Joseph KB Matovu, Saman Zamani, Rhoda K. Wanyenze.

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
