## [Decision Letter · Decision Letter 0]

PGPH-D-24-02391

Impact of layered behavioral, socio-economic and school-based interventions on selected behavioral and biomarker indicators among adolescent girls and young women in Uganda

Dear Dr. Matovu,

Thank you for submitting your manuscript to PLOS Global Public Health. After careful consideration, we feel that it has merit but does not fully meet PLOS Global Public Health’s publication criteria as it currently stands. Therefore, we invite you to submit a revised version of the manuscript that addresses the points raised during the review process.

We look forward to receiving your revised manuscript.

Kind regards,

Adriana Biney

Academic Editor

Journal Requirements:

Additional Editor Comments (if provided):

Reviewers' comments:

Reviewer's Responses to Questions

**Comments to the Author**

1. Does this manuscript meet PLOS Global Public Health’s publication criteria ? Is the manuscript technically sound, and do the data support the conclusions? The manuscript must describe methodologically and ethically rigorous research with conclusions that are appropriately drawn based on the data presented.

Reviewer #1: Yes

Reviewer #2: Yes

2. Has the statistical analysis been performed appropriately and rigorously?

Reviewer #1: Yes

Reviewer #2: Yes

3. Have the authors made all data underlying the findings in their manuscript fully available (please refer to the Data Availability Statement at the start of the manuscript PDF file)?

Reviewer #1: Yes

Reviewer #2: No

4. Is the manuscript presented in an intelligible fashion and written in standard English?

Reviewer #1: Yes

Reviewer #2: Yes

5. Review Comments to the Author

Reviewer #1: General comments

The study investigated the impact of layered behavioral, socio-economic and school-based interventions on selected behavioral and biomarker indicators among adolescent girls and young women in Uganda. A comprehensive background to the problem has been given. It suggests that whereas adolescent girls and young women are at an elevated risk of HIV infection globally, East and Southern African countries continue to be more vulnerable, including Uganda. Factors such as inequalities, discrimination and poverty deny many women and adolescent girls’ economic autonomy,

deprive them of control over their sexual lives, and expose them to ongoing HIV risk. Again, other identified drivers for HIV infection include intimate partner violence where women are unable to negotiate for safe sex with the abusive partner, child sexual abuse, and early sex debut which also contributes to teenage pregnancy. The study further urges that exposing adolescent girls and young women to multiple HIV prevention interventions such as cash transfer schemes or education subsidies, skills training programs and linking comprehensive sexuality education with sexual and reproductive health services, etc. have the potential of reducing the risks to HIV infection by decreasing the involvement in risky sexual behaviour and decreasing teenage motherhood. The study is relevant and has a clearly stated objective. The write up is good except for a few editorial issues as outlined below.

MAJOR ISSUES

Introduction

a. The background is well stated. However, the authors need to provide a brief description of why the study is important. What its public health significance?

Methodology

a. Line 23, pg. 4: It is stated that ‘‘Based on the program’s Theory of Change (Figure 2)…’’. The authors introduce to the readers a theory the study used. However, it would be good if the authors talk about briefly what the theory of change is about and its core principles before showing us how it was applied in their study.

b. On data analysis section, it would be helpful if the authors indicate first the kind of the analytical methods used in the study (statistical tests) before explaining the steps taken in the analysis. For example in the findings tables 5, 7, and 9, confidence intervals are stated which can be associated with specific types of statistical analyses. Again, they can state in a sentence or two the justification why those methods were chosen.

Discussion

a. Even though the authors showed how the theory of change was used for AGYW interventions, the discussions it is not shown how the findings are related to the theory.

b. The discussion of findings needs strengthening. Although the authors do well to explain what their findings mean in the context of their research, more is needed with regards to comparing the current findings with previous research. Additional literature on the current four used in the discussion is necessary.

c. Only one recommendation is provided even though important findings are identified which can influence recommendations for policy, practice, and research. This has to be improved. Some of such recommendations are stated in the discussion of findings and hence need to be teased out and brought to the rightful place. Again, since this is a crucial topic that has been studied, it would be good to suggest the bodies that will need to enforce the recommendations provided for the study to become relevant and contribute to the wellbeing of the participants. For example, is there a way the government, especially targeted ministries like the Ministry of health, policy makers, funders, schools, health care professionals, and other partners can come in to support and how?

MINOR ISSUES

Abstract

a. Line 21-22, pg. ii: The abstract is good. However, authors need to provide absolute numbers or percentages/proportions for some statements in the findings section for a clear picture. For example, half of the AGYW were in school (what would the readers consider as half in relation to your study population); about two-thirds were single/never married.

b. Line 32, pg. ii: ‘‘… HIV syphilis was lower…’’ Check out for these words and correct accordingly

Methodology

a. Line 29-32, pg. 10: the authors seem to state their objectives for the study in the data analysis section. If yes, then these need to be clearly stated in the background of the study and briefly mentioned here for reinforcement.

Discussion

a. Line 13, pg. 19 ‘‘ b) the period of exposure may have too short to allow for behavior change to occur’’ add the word ‘‘been’’ between ‘‘have and short’’.

Reviewer #2: Title: Impact of layered behavioral, socio-economic and school-based interventions on selected behavioral and biomarker indicators among adolescent girls and young women in Uganda

Hoever the title is somehow different from the stated objective of "This paper reports the findings of an impact evaluation of the impact of behavioral, socio-structural and educational interventions targeting in- and out-of-school AGYW on the triple burden of HIV, teenage pregnancy and intimate partner violence among AGYW in Uganda" stated in from line 34 page 2 to lines 2-3 page 3.

Abstract:

With a word count of 533 is well above the maximum 300 words allowed.

Abstract not expectd to have subsections acording to guidelines given.

Introduction:

This statement in lines 9-10 ".....HIV prevalence among adolescent girls (aged 12-19 years) increased from 4.2% to 5.7% compared to an increase from 1.8% to 4.5% among their male counterparts" contradicts what is stated in lines 6-7 "...15% of all new infections occur among adolescents aged 10–19 years, with 83% of adolescent infections occurring in girls."

Methods;

Lines 31-33 the statatement "In the 2023 survey specifically, data were also collected from the non-intervention villages and schools to aid in the computation of the difference-in-difference estimates during the assessment of the impact of the interventions on the behavioral and biomarker indicators (see ‘Measurement of variables’ below)" does it imply that in the 2018 survey data were not collected from non-intervention areas.

Discussion:

In lines 19-20 & 28-29 the authors make reference to "qualitative interviews" that were not presented in the results section.

The study period includes the period 2020-2023 when there was COVID 19 pandemic disruptions yet the authors do not mention its effect in the discussion/limitations.

6. PLOS authors have the option to publish the peer review history of their article (what does this mean? ). If published, this will include your full peer review and any attached files.

**Do you want your identity to be public for this peer review?** For information about this choice, including consent withdrawal, please see our Privacy Policy .

Reviewer #1: No

Reviewer #2: No

---

## [Decision Letter · Decision Letter 1]

PGPH-D-24-02391R1

Impact of layered behavioral, socio-economic and school-based interventions on selected behavioral and biomarker indicators among adolescent girls and young women in Uganda

Dear Dr. Matovu,

Thank you for submitting your manuscript to PLOS Global Public Health. After careful consideration, we feel that it has merit but does not fully meet PLOS Global Public Health’s publication criteria as it currently stands. Therefore, we invite you to submit a revised version of the manuscript that addresses the points raised during the review process.

We look forward to receiving your revised manuscript.

Kind regards,

Adriana Biney

Academic Editor

Journal Requirements:

Additional Editor Comments (if provided):

Reviewers' comments:

Reviewer's Responses to Questions

**Comments to the Author**

1. If the authors have adequately addressed your comments raised in a previous round of review and you feel that this manuscript is now acceptable for publication, you may indicate that here to bypass the “Comments to the Author” section, enter your conflict of interest statement in the “Confidential to Editor” section, and submit your "Accept" recommendation.

Reviewer #2: All comments have been addressed

2. Does this manuscript meet PLOS Global Public Health’s publication criteria ? Is the manuscript technically sound, and do the data support the conclusions? The manuscript must describe methodologically and ethically rigorous research with conclusions that are appropriately drawn based on the data presented.

Reviewer #2: Yes

3. Has the statistical analysis been performed appropriately and rigorously?

Reviewer #2: Yes

4. Have the authors made all data underlying the findings in their manuscript fully available (please refer to the Data Availability Statement at the start of the manuscript PDF file)?

Reviewer #2: No

5. Is the manuscript presented in an intelligible fashion and written in standard English?

Reviewer #2: Yes

6. Review Comments to the Author

Reviewer #2: General comments:

The study covered the years 2018 - 2023 and this include the COVID19 pandemic period 2020 - 2022 marked with lockdowns and various restrictions yet the authors have conveniently avoided discussing its effect on the interventions an results.

The survey in 2018 was conducted in 20 districts while the one in 2023 was carried out in 14 districts. An explanation ought to e given as to why the 6 districts were not included in 2023.

Specific comments:

Abstract - Line 32 the phrase "HIV Syphillis" needs to be corrected.

Introduction - phrase in lines 9 - 10 appear to be contradictory to what is stated in lines 6 -7.

7. PLOS authors have the option to publish the peer review history of their article (what does this mean? ). If published, this will include your full peer review and any attached files.

**Do you want your identity to be public for this peer review?** For information about this choice, including consent withdrawal, please see our Privacy Policy .

Reviewer #2: No

---

## [Editor Report · Decision Letter 2]

Impact of layered behavioral, socio-economic and school-based interventions on selected behavioral and biomarker indicators among adolescent girls and young women in Uganda

PGPH-D-24-02391R2

Dear Dr. Matovu,

We are pleased to inform you that your manuscript 'Impact of layered behavioral, socio-economic and school-based interventions on selected behavioral and biomarker indicators among adolescent girls and young women in Uganda' has been provisionally accepted for publication in PLOS Global Public Health.

Best regards,

Adriana Biney

Academic Editor